# Natural scene statistics predict how humans pool information across space in surface tilt estimation

**Seha Kim** [1]*, **Johannes Burge** [1,2,3]

**1** Department of Psychology, University of Pennsylvania, Philadelphia, Pennsylvania, United States of America, **2** Neuroscience Graduate Group, University of Pennsylvania, Philadelphia, Pennsylvania, United States of America, **3** Bioengineering Graduate Group, University of Pennsylvania, Philadelphia, Pennsylvania, United States of America

* sehakim@upenn.edu

**Data Availability Statement:** Data files from the human experiment and modeling efforts are available from the OSF database (https://osf.io/s9fhj).

**Funding:** This work was supported by NIH grant R01-EY028571 from the National Eye Institute &

## Abstract

Visual systems estimate the three-dimensional (3D) structure of scenes from information in two-dimensional (2D) retinal images. Visual systems use multiple sources of information to improve the accuracy of these estimates, including statistical knowledge of the probable spatial arrangements of natural scenes. Here, we examine how 3D surface tilts are spatially related in real-world scenes, and show that humans pool information across space when estimating surface tilt in accordance with these spatial relationships. We develop a hierarchical model of surface tilt estimation that is grounded in the statistics of tilt in natural scenes and images. The model computes a global tilt estimate by pooling local tilt estimates within an adaptive spatial neighborhood. The spatial neighborhood in which local estimates are pooled changes according to the value of the local estimate at a target location. The hierarchical model provides more accurate estimates of groundtruth tilt in natural scenes and provides a better account of human performance than the local estimates. Taken together, the results imply that the human visual system pools information about surface tilt across space in accordance with natural scene statistics.

## Author summary

Visual systems estimate three-dimensional (3D) properties of scenes from two-dimensional images on the retinas. To solve this difficult problem as accurately as possible, visual systems use many available sources of information, including information about how the 3D properties of the world are spatially arranged. This manuscript reports a systematic analysis of 3D surface tilt in natural scenes, a model of surface tilt estimation that makes use of these scene statistics, and human psychophysical data on the estimation of surface tilt from natural images. The results show that the regularities present in the natural environment predict both how to maximize the accuracy of tilt estimation and how to maximize the prediction of human performance. This work contributes to a growing line of

Office of Behavioral and Social Science Research and NIH grant R01-EY011747 from the National Eye Institute (to JB). The funders had no role in study design, data collection and analysis, decision to publish, or preparation of the manuscript.

**Competing interests:** The authors have declared that no competing interests exist.

work that establishes links between rigorous measurements of natural scenes and the function of sensory and perceptual systems.

## Introduction

Estimating three-dimensional (3D) surface orientation from two-dimensional (2D) retinal images is one of the most critical functions of human vision [1]. To determine whether a surface can be walked on or used to hang a picture, its 3D orientation must be accurately estimated. Laboratory studies have typically examined the estimation of 3D orientation with isolated planar surfaces that are textured with simple patterns [2–13]. However, the estimation problem in the real world is often more complex than situations that are commonly studied in the lab. Real-world surfaces have varied textures and complicated geometries [14–16]. To understand how the estimation of 3D surface orientation works in the real world, it can be useful to study performance with stimuli that are as natural as possible. In images of natural scenes, statistical information about local image cues and the most probable three-dimensional spatial contexts can both provide useful information. In a variety of visual tasks with artificial stimuli, the spatial (i.e., global) context surrounding a given set of local cues can affect local percepts [17–21]; well-known examples include the simultaneous contrast illusion, the simultaneous color contrast illusion, and the slant contrast illusion [22,23]. But it is not always clear how to account for these effects. The computer vision literature frequently models the use of spatial context, but infrequently provides insights into the computations that may underlie human performance [24]. In the vision literature, there have been attempts to develop quantitative models that capture the impact of spatial context on human perception, but the stimuli that these models apply to are often rather artificial [6,25]. There have also been multiple demonstrations that global context influences human perception of surface orientation in real-world 3D scenes [26], but these studies typically do not provide quantitative models that account for human performance.

Surface orientation is typically parameterized by slant and tilt [27]. Slant is the amount by which a surface is rotated away from an observer. Tilt is the direction in which the surface is rotated (Fig 1). In this paper, we examine how humans incorporate spatial context to estimate 3D surface orientation in real-world scenes. To use spatial context, visual systems must integrate (or pool) information across space. To model this pooling process, we propose a two-stage hierarchical model.

In the first stage, 3D tilt is estimated at each of multiple spatial locations, using the joint statistics relating local cues in images to groundtruth tilts in natural scenes [15,16]. At each spatial location, the estimate is Bayes-optimal given measurements of three local image cues: local gradients of luminance, texture, and disparity. This estimate is referred to as a *local-model estimate* because it is based on local cues only. We have previously found that this local model provides a nice account of human tilt estimation performance, including a pronounced bias towards the cardinal tilts [16]. This previous finding is consistent with the hypothesis that the human visual system incorporates the prior probability that cardinal tilts ($0°$, $90°$, $180°$, $270°$) are more common than other tilts in natural scenes (see below) [15,16,28,29]. In the second stage, the local estimates are pooled across multiple spatial locations to obtain a *global-model estimate*. The global pooling rules are motivated by how tilts are spatially related in real-world scenes. The current modeling efforts extend previous work by capitalizing upon how tilts in real-world scenes are related across space [16].

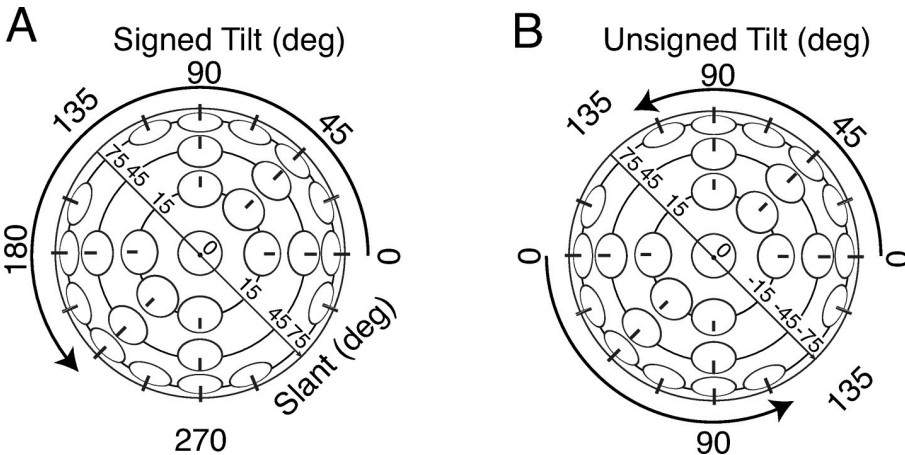

**Fig 1. 3D surface orientation is fully described by slant and tilt.** Slant is the angle indicating how much a surface is rotated out of the fronto parallel plane. Tilt is the direction of slant, as quantified by the angle between the x-axis in the frontoparallel plane and the surface normal projected into the frontoparallel plane. **A** Signed tilt, defined on [0˚,360˚), and unsigned slant. **B** Unsigned tilt, defined on [0˚,180˚), and signed slant.

We examine the ability of the global model i) to estimate tilt in real-world scenes and ii) to predict human estimation of tilt in those same scenes. We find that the global pooling model provides more accurate estimates of tilt and better predictions of human performance than the local model. Additionally, we find that the size of pooling region that optimizes estimation performance is approximately the same as the size of the pooling region that optimizes the model predictions of human performance. The results suggest that the human visual system pools information over the spatial region that optimizes tilt estimation performance in natural scenes.

## Results

### Natural scene statistics of tilt

In images of natural scenes, 3D surface orientations corresponding to nearby image locations are correlated. This is because natural scenes tend to be dominated by continuous surfaces, and surface discontinuities tend to be comparatively rare [30]. Visual systems that internalize and properly use the statistics governing these spatial relationships in natural scenes will outperform visual systems that do not. These scene statistics motivate the global pooling rules that are the primary focus of the paper.

To determine how surface tilts in real-world scenes are related across space, we analyzed a recently published database of natural stereo-images with precisely co-registered time-of-flight laser-based distance measurements at each pixel [15] (Fig 2A). Groundtruth surface tilt was computed from the distance measurements at each pixel (Fig 2B; see Methods). The distribution of surface tilts in natural scenes—the prior probability distribution over tilt—has pronounced peaks at the cardinal tilts (Fig 2C).

The rules for pooling information across space that maximize estimation performance depend critically on how the variable to be estimated (e.g., tilt) is correlated across space. Unfortunately, the correlation of circular (i.e., angular) variables is notoriously unstable when the variables are highly dispersed, and it is known that local tilt in natural scenes is a highly dispersed circular variable [16]. This fact makes it difficult to precisely link measured scene statistics to optimal pooling rules. Thus, rather than to quantitatively specify the optimal pooling

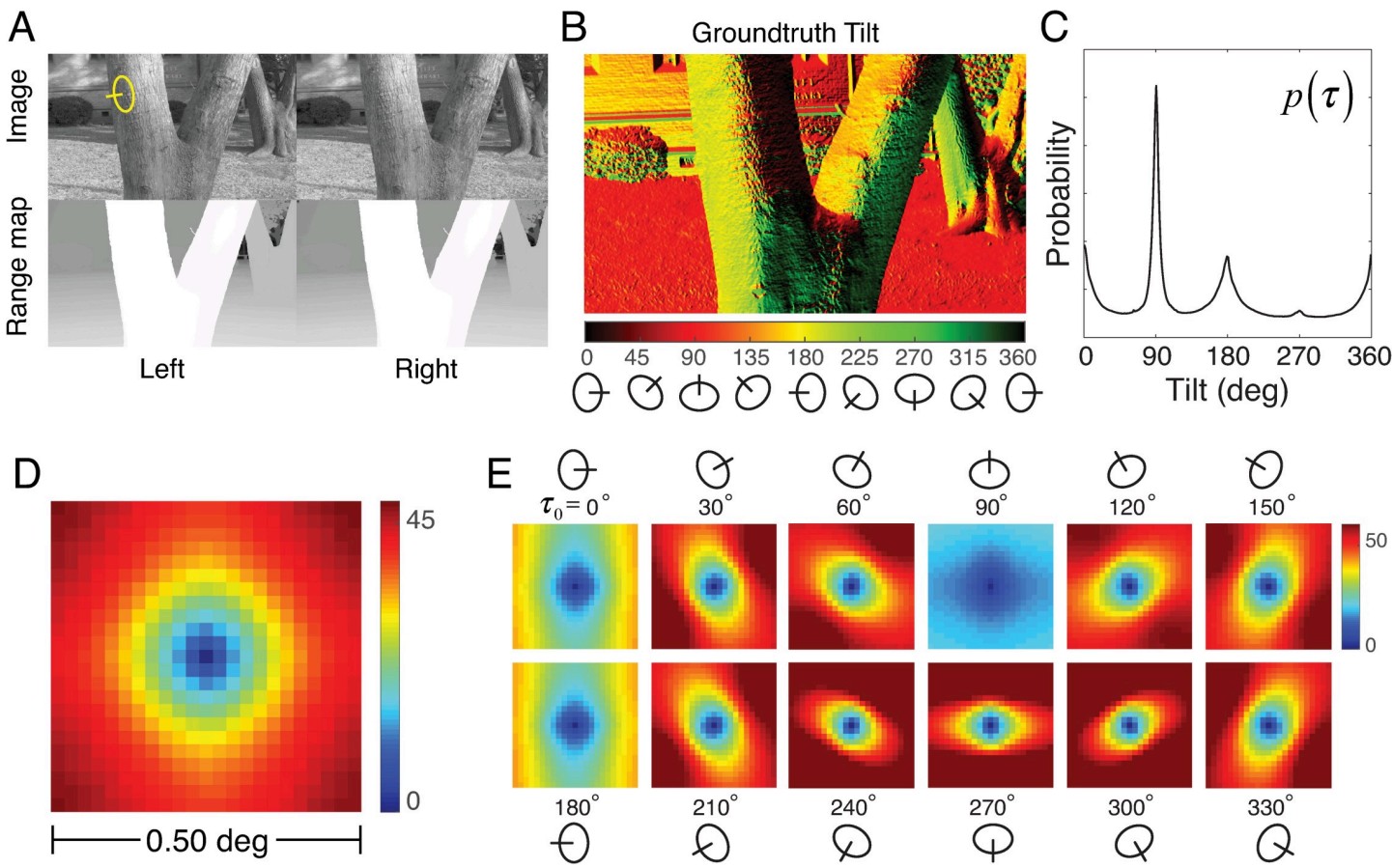

**Fig 2. Spatial statistics of tilt in natural scenes. A** Stereo images and stereo distance maps of real-world scenes. The distance data is co-registered to the image data at each pixel. **B** Groundtruth tilt corresponding to the image in A. Groundtruth tilt at each pixel is computed directly from the data in the distance maps. **C** Prior distribution of groundtruth tilt, computed from 600 million groundtruth tilt samples in the natural scene database. **D** Mean absolute tilt difference from the center target tilt as a function of spatial location. The color represents the tilt difference across all pixels in all images in the natural scene database. **E** Mean absolute tilt difference conditioned on the groundtruth tilt at the target location.

rules from first principles, we used the available natural scene statistics to motivate an exploration of plausible pooling rules.

As an alternative to spatial correlation, we computed the *mean absolute tilt difference* $E[|\tau_i - \tau_j|]$ between tilts corresponding to image locations $i$ and $j$. (Note that, in the main text, we use $\tau_i - \tau_j$ as notational shorthand for the circular distance between two angles; see Methods). Fig 2D shows the mean absolute tilt differences $E[|\tau_i - \tau_0|]$ in natural scenes for all possible spatial relationships in a spatial neighborhood surrounding the target tilt $\tau_0$ at the center of an image region. Unsurprisingly, the tilt differences increase systematically as spatial distance increases; the iso-difference contours are approximately circular. This finding suggests that pooling local tilt estimates within a circular neighborhood centered on the target tilt will increase estimation accuracy (see below).

Richer statistical structure is revealed when the tilt differences are conditioned on the central target tilt (i.e., $E[|\tau_i - \tau_0| |\tau_0]$). The size and shape of the neighborhood within which tilts are most similar change dramatically with the target tilt; the iso-difference contours are approximately elliptical (Fig 2E). For example, if the tilt at the center of a spatial area is equal to 0º (e.g., the side of a tree trunk), tilts at spatial locations above and below the center are more likely to be similar to the target tilt than tilts to the left and right. Thus, for a central target tilt

of 0°, a vertically elongated pooling region may be appropriate. For a central target tilt of 90˚ (e.g., the ground plane), spatial locations to the left and right of the center are most likely to be similar to the target tilt, and neighboring tilts are likely to be similar over a larger area. More generally, the statistics suggest that the pooling region should be elongated in a direction that is orthogonal to the tilt direction. Visual systems that use local estimates, and pool them adaptively in a manner that is consistent with these statistical regularities, have the potential to outperform visual systems that pool local estimates with fixed (non-adaptive) neighborhoods. Before developing specific models that will investigate these ideas, we describe a psychophysical experiment that we performed to determine how humans estimate tilt in natural scenes.

## Psychophysical experiment

To test how human observers estimate 3D tilt in natural scenes, we performed a psychophysical experiment. The scenes were obtained from the same database that we used to analyze the natural scene statistics [15] (see above). Two sets of 3600 scene locations were randomly sampled from the database and were used in two experiments under specified constraints. Locations were sampled only if the surfaces at those locations were within a specified range of distances, slants, and image contrasts. The constraints ensured that the image cues were measurable by the human visual system, that the task was well defined, and that the stimuli could be presented without artifacts on our display system (see Methods).

Scene locations were displayed on a large stereo-projection system at a 3m viewing distance (Fig 3A). The display system creates retinal images that provide a close approximation to the retinal images and stereo-viewing geometry that viewing the original scene would have created [16]. On each trial, the observer viewed a scene location through a 3º diameter circular stereoscopic window. The task was to estimate the 3D tilt of the surface at the center of each stereoscopic window. Observers indicated their estimates with a mouse-controlled graphical probe, a white circle surrounding the window with three radially protruding tick marks that were ninety degrees apart (Fig 3B). Observers aligned the middle tick mark of the probe with the

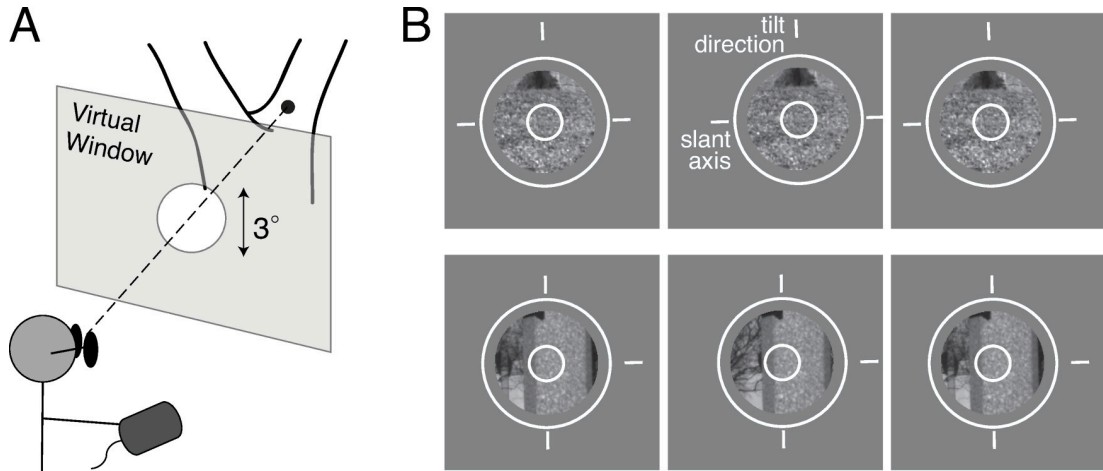

**Fig 3. Human tilt estimation experiment. A** Human observers binocularly viewed real-world scenes through a circular aperture with a 3º diameter that was positioned stereoscopically in front of the scene. **B** Example of stimuli. Left-eye, right-eye, and left-eye images (for both uncrossed and crossed fusion). The patches are surrounded by a graphical probe (white circle and three tick marks). Observers rotated the probe to align the middle tick mark with the perceived tilt direction for the surface at the very center of the window; note that when the middle tick mark is aligned with the perceived tilt direction, the other two tick marks are aligned with the perceived slant axis.

perceived tilt of the surface at the center of the window. Human tilt estimation performance is presented alongside modeling results in subsequent sections.

The two experiments were used as an internal check on the consistency of the results. The first experiment sampled surfaces with a larger range of slants than the second experiment. Experiment 1 used stimuli with slants ranging between 30º and 75º. Experiment 2 used stimuli with slants between 30º and 60º. Progressively more steeply slanted surfaces project to progressively smaller regions of the image, assuming the surfaces are equal in size. This fact about projective geometry should make groundtruth tilt more difficult to estimate, especially when the surface is embedded in a cluttered environment, which is often the case in natural scenes. The results between the two experiments differed by a modest amount, but in the expected direction. Performance was better when samples with very high slants were excluded (Experiment 2), but the performance differences with Experiment 1 were modest (see Discussion).

## Modeling

The proposed model of tilt estimation has two hierarchical processing stages. In the first stage, local estimates are computed from image cues that are extracted from natural images. In the second stage, global estimates are obtained by pooling the local estimates within a spatial neighborhood centered on a target location; the global pooling rules are motivated by our statistical analyses (c.f. Fig 2D and 2E) of how tilts are spatially related in natural scenes. These two processing stages are described in order.

The modeling effort described here builds on previous work in two ways [16]. The primary development is in how the current model makes use of spatial context; local estimates are pooled based on the statistics of surface orientation in real-world scenes. A secondary development is that the local estimates are now of *signed* tilt (i.e., both tilt magnitude and sign) rather than of only *unsigned* tilt (i.e., tilt magnitude; Fig 1). These developments allow us to investigate the manner in which humans pool estimates of signed tilt across space.

**Local signed tilt estimation.** The first stage of the hierarchical model estimates local tilt, relying on the statistics relating local image cues and surface tilt in natural scenes. The scene statistics are compiled from hundreds of millions of samples from the previously mentioned database of natural scenes [15]. The local estimation stage is based closely on a previously published local model that predicted many of the successes and failures of human tilt estimation in natural scenes [15,16]. However, the previous model had a shortcoming; it provided estimates only of unsigned tilt. The local model proposed here provides estimates of both tilt magnitude and sign (i.e., signed tilt).

The local model first computes the estimate of unsigned tilt given three unsigned image cues: local luminance, texture, and disparity gradients. The estimate of unsigned tilt $\hat{\tau}_u$ specifies the tilt *magnitude* and is equal to the mean of the posterior over unsigned tilt given the unsigned image cues,

$$\hat{\tau}_u = E[\tau_u | \mathbf{C}_u] = \sum_{\tau_u} \tau_u p(\tau_u | \mathbf{C}_u), \tag{1}$$

where $\tau_u$ is the unsigned groundtruth tilt, and $\mathbf{C}_u$ is a vector of three unsigned cue values (Fig 4A, cube). (Note that the expression for the posterior mean in Eq 1 is used as notational shorthand for the mean of a probability distribution over a circular variable; see Methods. Also note that the posterior probability distribution incorporates the prior probability distribution of tilt in the natural scene database.) The posterior mean is equivalent to the Bayes-optimal estimate assuming the circular analog to a squared-error cost function (see Methods).

The model then obtains the estimate of *tilt sign* $\text{sgn}(\hat{\tau}_s)$ by computing the mean of the posterior over signed tilt given the signed disparity cue (Fig 4A, bar), which is the only cue

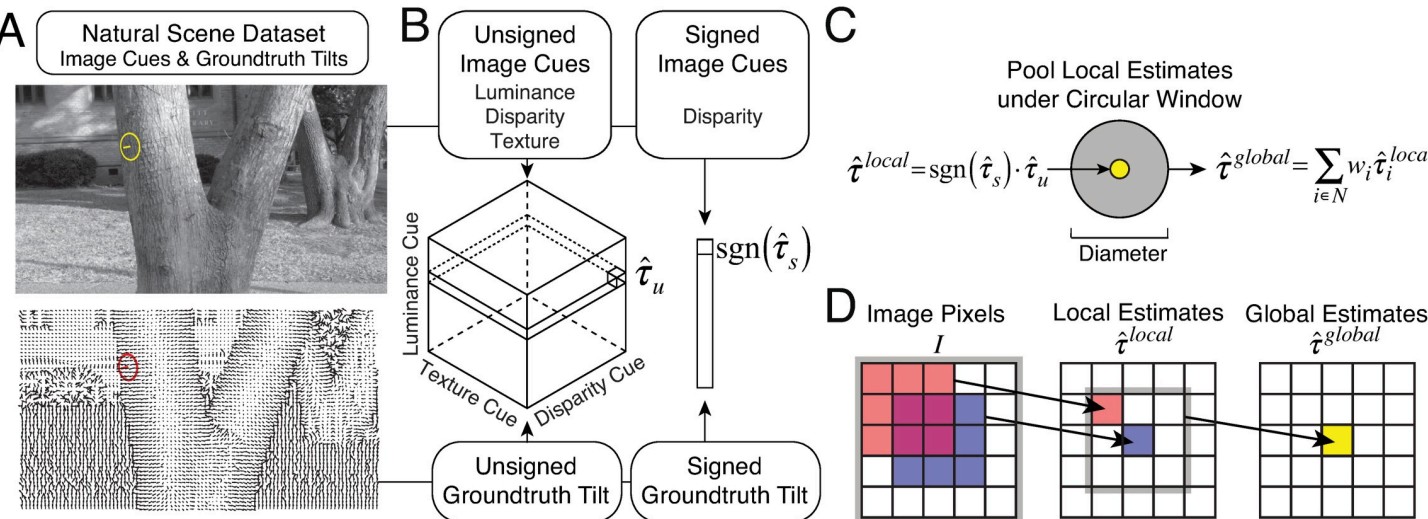

**Fig 4. Constructing local and global models of tilt estimation.** **A** Image cues and groundtruth tilt in natural scenes. Image cues are derived directly from photographic stereo images (top). Groundtruth tilt at each pixel is computed directly from the range data (cf. Fig 2A). Here, groundtruth tilt is depicted with local surface normals instead of a colormap (cf. Fig 2B). **B** The local model estimates tilt based on local image cues. Local estimates are obtained via lookup tables that store conditional means (i.e., posterior means) given all possible combinations of three quantized unsigned image cue values (i.e., $64^3$ unique cue combinations), and one quantized signed image cue value (i.e., 64 unique cue values), as computed from the natural image database. We have previously verified that quantizing the cue values is not a primary limiting factor on the performance of the model [16]. **C** Pooling local estimates in a spatial pooling region centered on a target location. **D** Each global estimate is obtained by pooling local estimates over a spatial neighborhood. Each local estimate is obtained by combining cues that are computed from multiple pixels in the image. Note that the area of the image that contributes to the global estimate is slightly larger than the purported area of the global pooling region, because each local estimate is computed from image gradients across an image region with non-zero spatial extent.

providing reliable information about tilt sign. Specifically,

$$\text{sgn}(\hat{\tau}_s) = \text{sgn}(E[\tau_s|\mathbf{C}_s]) = \text{sgn}(\sum_{\tau_s} \tau_s p(\tau_s|\mathbf{C}_s)), \qquad (2)$$

where $\tau_s$ is the signed groundtruth tilt, and $\mathbf{C}_s$ is the signed disparity cue (Fig 4A bar). The final local estimate of *signed* tilt is obtained by multiplying the estimate of tilt magnitude by the estimate of tilt sign

$$\hat{\tau}^{local} = \hat{\tau}_u \cdot \text{sgn}(\hat{\tau}_s), \qquad (3)$$

An alternative approach would be to compute the posterior mean over signed tilt given all three image cues. Doing so, however, produces larger errors. We favor the more accurate method. Does this local model differ substantially from a model that computes optimal tilt estimates from disparity information alone? A previously published analysis showed that when the other two cues substantially differ from each other, they change the disparity-alone estimate by a small amount. But when the other two cues differ from disparity and approximately agree with each other, the model estimate differs considerably from the disparity-alone estimate [15].

The local estimates are used as input to the second stage of the hierarchical model. Performance of the local model will be compared to the performance of models that use global pooling in subsequent sections.

**Global tilt estimation.** The second stage of the hierarchical model pools local estimates to improve performance. The pooling rules that we investigate are based on the statistical properties of tilt in natural scenes. We have shown that groundtruth tilt signals exhibit statistical regularities across space (cf. Fig 2D and 2E). Under these conditions, pooling local estimates has the potential to average out noise and improve performance [31]. But the benefit of averaging out noise by pooling must be balanced against the risk of averaging over groundtruth signals

that are changing across space. Considering two extremes helps drive the point home. On one extreme, if local groundtruth signals are perfectly correlated across space, then the optimal pooling rule is to average all local estimates across the largest possible area. On the other extreme, if local groundtruth signals are perfectly uncorrelated across space (i.e., random), any spatial pooling at all degrades performance. Thus, to realize performance improvements, the pooling rules must be well matched to the governing statistics. If all local estimates are equally reliable, for example, the optimal pooling area should be determined by the spatial correlation of the groundtruth signals.

The global pooling models proposed here compute a global tilt estimate at a given target location from a weighted sum of the local estimates in a spatial neighborhood centered on the target location (Fig 4B). The specific weights and the neighborhood together represent the pooling rule. Specifically, the global estimate is given by

$$\hat{\tau}^{global} = \sum_{i \in N} w_i \hat{\tau}_i^{local},$$ (4)

where $N$ is the spatial neighborhood, and $w_i$ is the weight for each local tilt estimate within the neighborhood [32–34]. (Note that Eq 4 is notational shorthand for the weighted circular mean; see Methods). Pooling local estimates causes image information (i.e., pixels) from an area larger than the pooling region to contribute to each global estimate (Fig 4D). However, the area from which the local estimates are computed tends to be small relative to the size of the global pooling region; that is, the global pooling regions tend to be quite large relative to the image area from which the local cues are estimated.

We examine the performance of global pooling relative to the local model (Fig 5A) in the context of two global pooling strategies: *fixed circular pooling* and *adaptive elliptical pooling* (Fig 5BC). The fixed circular pooling model uses the same pooling area regardless of the tilt at the target location in the center of the pooling region (Fig 5B). The adaptive elliptical pooling model changes the pooling region with the target tilt (Fig 5C). Each of these pooling strategies is motivated by the natural scene statistics shown in Fig 2D and Fig 2E, respectively, and is discussed in more detail below.

We quantify the performance of each model in two ways. First, for a given pooling strategy, we analyze how it changes the accuracy of groundtruth tilt estimation. Second, we analyze how well a given pooling strategy accounts for human performance. If the human visual system uses global context to estimate tilt, then human responses should be better predicted by a global model that uses spatial context than by a local model that uses only local image cues. By comparing the neighborhood sizes that optimize groundtruth tilt estimation and that maximize the prediction of human performance, we gain insight into the pooling strategy that humans use when estimating tilt in natural scenes.

**Fixed circular pooling: Modeling results and human performance.** We start by considering a model with a fixed circular pooling. A fixed circular pooling region is centered on the target, and it equally weights each local estimate in the pooling region (i.e., $w_i = w_j$ for all $i$ and $j$; Fig 4C). The circular shape of the pooling region is motivated by the circular shape of the iso-similarity contours in natural tilt statistics (see Fig 2D). For this model to be optimal, two conditions must be satisfied, assuming zero noise correlations. The first condition is that all groundtruth tilts within the pooling region must have the same value. The second condition is that local estimates, regardless of their value, must provide equally reliable information about the groundtruth tilt that gave rise to the estimate. Although neither condition can be strictly true, there is some empirical justification for each. First, groundtruth tilts within a sufficiently small circular area tend to be quite similar (Fig 2D). Assuming that all tilts are equal within the pooling region is therefore not an unreasonable approximation, provided the pooling region is

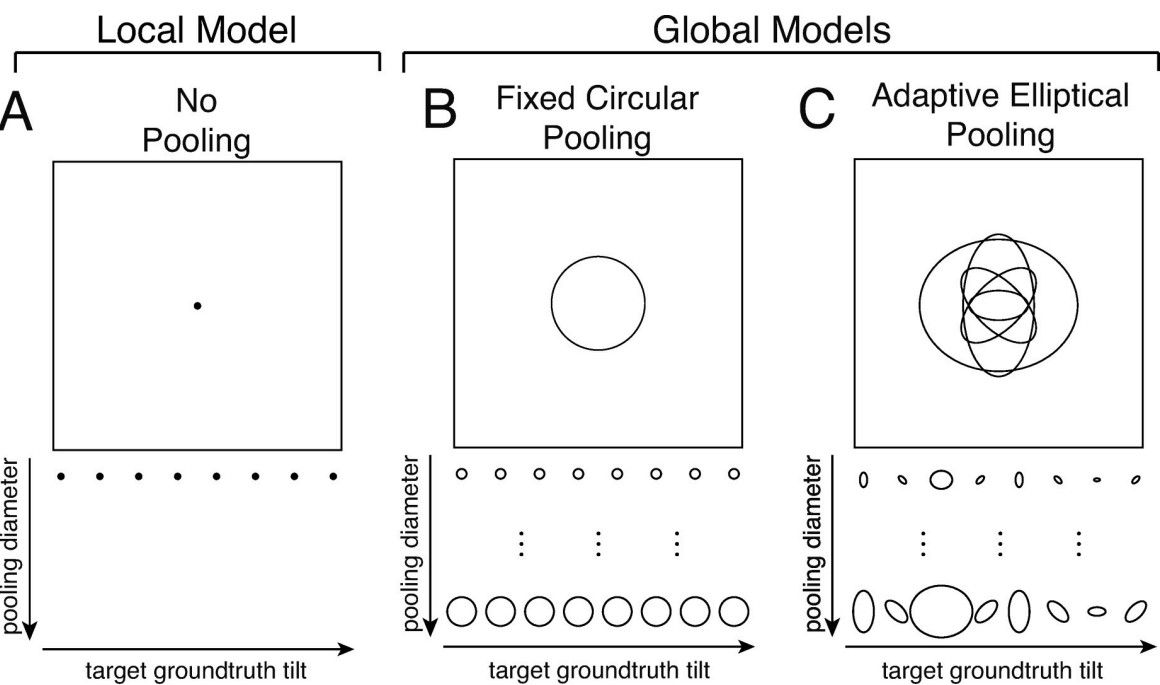

**Fig 5. Local and global models for tilt estimation. A** The local model obtains a local tilt estimate given three local image cues. **B** The fixed circular pooling model uses a circular pooling region with the same size for all target groundtruth tilts (cf. Fig 2D). **C** The adaptive elliptical pooling model uses an adaptive pooling region with a different size, aspect ratio, and orientation for each groundtruth tilt (cf. Fig 2E). As the average area of the adaptive elliptical pooling region changes, the relative area, orientation, and aspect ratio of the pooling regions are held fixed.

not too large. Second, probability distributions over groundtruth tilt given a local estimate with a particular value (obtained from the local model) $p(\tau|\hat{\tau}^{local})$ are approximately shift-invariant (Fig 6); each local estimate is thus an equally reliable predictor of groundtruth tilt

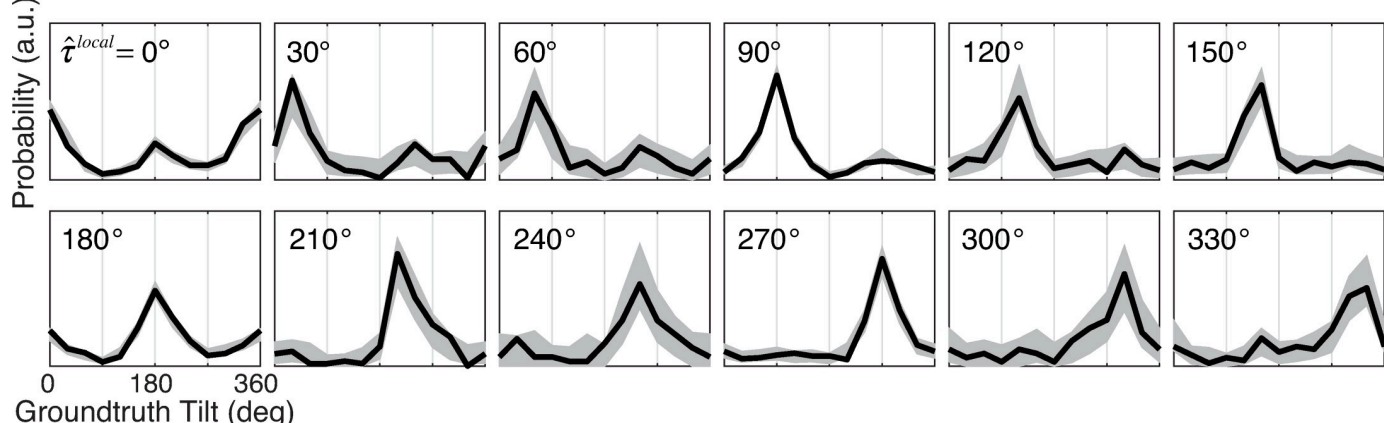

**Fig 6. Conditional distributions of groundtruth tilt given the value of the tilt estimate.** Each subplot shows the distribution of groundtruth tilt given a particular local estimate value $p(\tau|\hat{\tau}^{local})$. For example, the fifth subplot in the first row shows the distribution of groundtruth tilts given that the local tilt estimate had a value of $120°$ (i.e., $p(\tau|\hat{\tau}^{local} = 120°)$). The fact that the conditional distributions of groundtruth tilt are approximately shift-invariant indicates that each local tilt estimate, regardless of its value, provides approximately equally reliable information about groundtruth tilt. Gray regions represent 95% confidence intervals from Monte Carlo simulations of 1000 experimental datasets. Confidence intervals at non-cardinal tilts (e.g., $\hat{\tau}^{local} = 30°$, $60°$, $120°$, $150°$, etc.) are larger in part because the local model produces fewer non-cardinal tilt estimates, in keeping with the prior probability distribution over tilt, which has peaks at the cardinal tilts (e.g., $\tau = 0°$, $90°$, etc.; see Fig 2C).

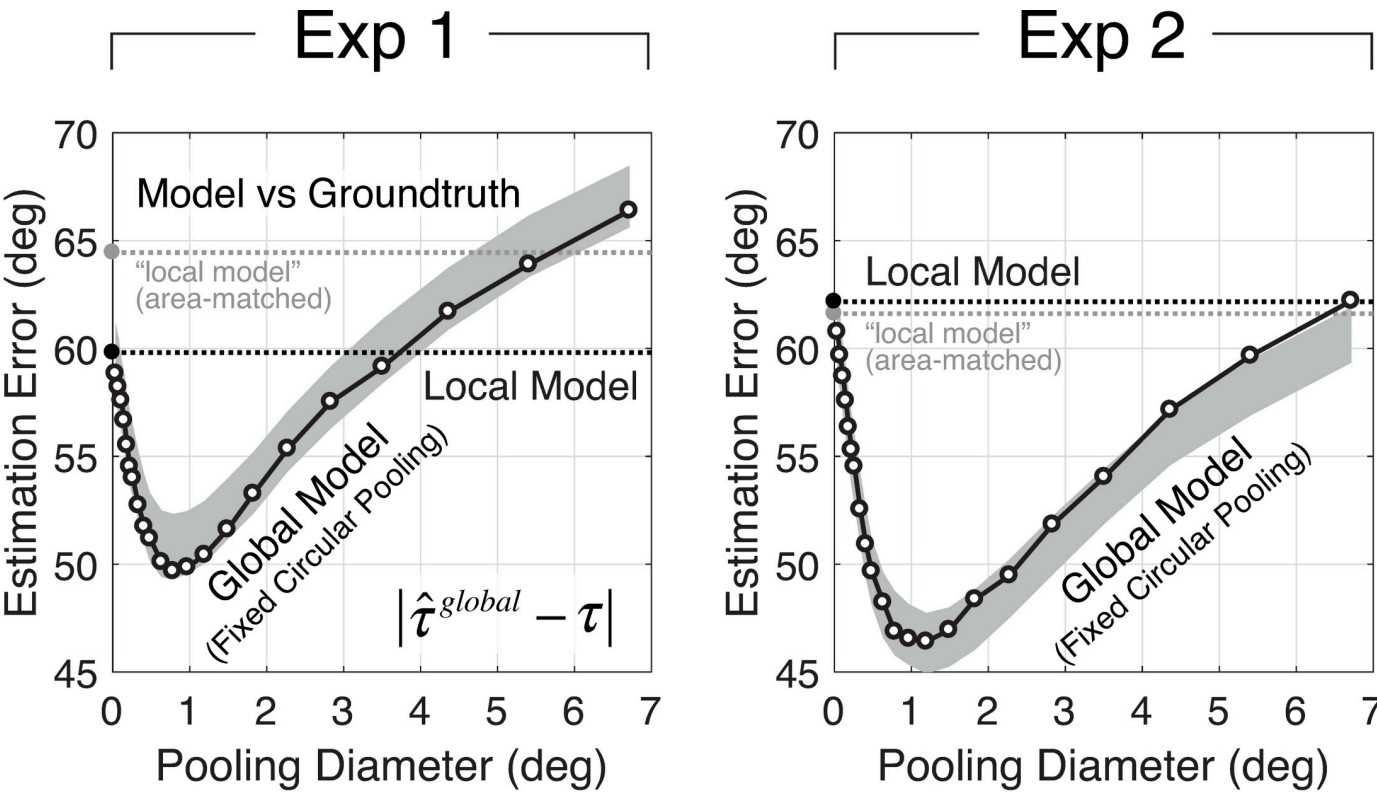

**Fig 7. Groundtruth tilt estimation error from the global model with fixed circular pooling.** Mean estimation error is plotted as a function of the diameter of pooling region. Mean estimation errors are computed across all tilts. The black dashed line indicates the mean estimation error for the local model; the local model does not pool local estimates and thus has a pooling diameter of 0º. The gray dashed line indicates the estimation error for a "local" model that computes the image cues from an area matched to that implicitly used by the best global model (see Discussion). Monte Carlo simulations on 1000 randomly sampled stimulus sets were used to obtain 95% confidence intervals on the mean estimation error (gray area). Data from Exp 1 and Exp 2 are shown in the left and right columns, respectively.

regardless of its value [16]. Pooling local tilt estimates with equal weights in a circular region is thus a reasonable starting point for investigating the degree to which spatial pooling can improve performance.

To analyze the fixed circular pooling model, we examined how performance changes as a function of the pooling region diameter. First, we determined the size of the pooling region that produces the best estimates of groundtruth tilt in natural scenes (Fig 7). Second, we determined the size of the pooling region that maximizes the model prediction of human estimates (Fig 8). We found that a pooling region of approximately 1.0º of visual angle is associated with the best performance.

To evaluate the model's groundtruth tilt estimation performance, we computed the *estimation error* between model estimates and groundtruth tilts across the entire stimulus ensemble used in the psychophysical experiment. The estimation error is the circular distance between the model estimate and groundtruth tilt. We express neighborhood size by the diameter of the pooling region. Mean estimation error across all stimuli is plotted as a function of the pooling diameter of the circular region (Fig 7). With fixed circular pooling, estimation error decreases as the pooling diameter increases until it reaches a critical pooling diameter that optimizes performance. The critical pooling diameter is approximately 1.0º. As pooling diameter increases further, estimation error begins to increase. Once pooling diameters exceed 3.5º, the global model fails to outperform the local model (Fig 7; black dashed line). These results show that, for a range of pooling diameters, the global model with fixed circular pooling provides more

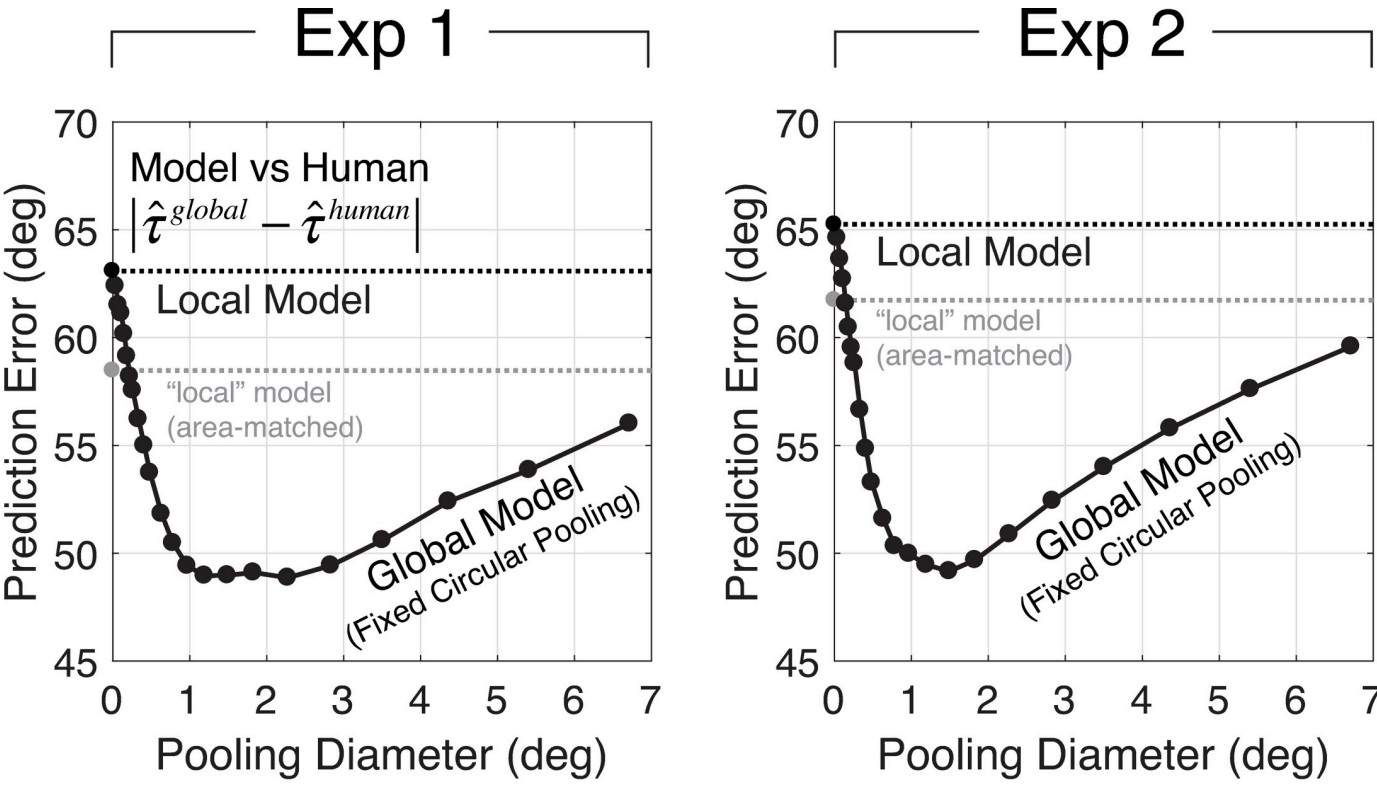

**Fig 8. Human prediction error from the global model with fixed circular pooling.** Mean prediction error is plotted as a function of the diameter of the pooling region. Mean prediction errors are computed across all tilts and human observers. The black dashed line indicates the mean prediction error for the local model. The gray dashed line indicates the prediction error for a "local" model that computes the image cues from an area matched to that implicitly used by the best global model (see Discussion). Data from Exp 1 and Exp 2 are shown in the left and right columns, respectively.

accurate estimates of tilt than does the local model. The overall benefit of global pooling is small (~10°), but it is robust. To ensure that this result is not an artifact of the experimental stimulus set, we analyzed estimation errors with a Monte Carlo simulation on 1000 sets of randomly sampled stimuli. To ensure that these results are not due simply to the fact that the global model uses information from a larger image area than the local model, we show that an area-matched "local" model underperforms this global model (see Discussion). The results show that global pooling consistently reduces estimation error; the performance improvements are not due to the particular sample of stimuli used in the psychophysical experiment.

To examine whether global pooling predicts human performance better than the local model, we computed the mean *prediction error* between the model estimates and the human estimates across the stimuli used in the psychophysical experiment. (The details of the human estimates are shown in S1 and S2 Figs). The prediction error on any given stimulus is the absolute circular distance between the model estimate and human estimate. The mean prediction error is the average prediction error across all stimuli and observers. Mean prediction error across all stimuli is plotted as a function of the diameter of the circular pooling region (Fig 8). Just as with estimation error, prediction error decreases as pooling diameter increases, until a critical diameter is reached. The pooling diameter that minimizes prediction error is between 1.0° and 2.0°. This diameter is similar to the diameter that minimizes estimation error. The same result holds for individual human observers; the pooling diameter that minimizes prediction error is between 1.0–1.5° for four of five human datasets (S3 Fig). This result suggests that the human visual system pools local estimates over an area that is sized to balance the benefits

(i.e., averaging out measurement noise) and the costs (i.e., pooling over irrelevant different groundtruth tilts) to maximize accuracy.

**Adaptive elliptical pooling: Modeling results and human performance.** Pooling local tilt estimates within a fixed circular neighborhood confers a performance benefit compared to no pooling at all. Our analyses of natural scene statistics show that the spatial neighborhood in which nearby tilts are most similar to the target tilt depends on the target tilt itself (see Fig 2E). These elliptical regions of similarity suggest that a strategy more sophisticated than fixed circular pooling may yield additional performance improvements. The adaptive elliptical pooling model pools local estimates within an elliptical neighborhood that changes adaptively with the target tilt. The orientation, aspect ratio, and relative size of the elliptical pooling regions were fit to and fixed by the scene statistics in Fig 2E (see Methods). The results of these fits are shown in Fig 9A.

To determine the performance of the adaptive elliptical pooling model, we varied the average size of the elliptical pooling regions while keeping the pattern of relative size, orientation, and aspect ratio fixed, and then computed groundtruth tilt estimation errors and human prediction errors. These errors were compared to the errors obtained with fixed circular pooling and the local model. However, direct comparison with fixed circular pooling is complicated by the relative size and shape changes associated with adaptive elliptical pooling. To address this problem, we defined the *equivalent diameter* of a given ellipse as the diameter of the circle $D = 2\sqrt{A/\pi}$ that has the same area $A$ as the ellipse. The *average equivalent diameter* $\bar{D} = 2\sqrt{\bar{A}/\pi}$ corresponds to the average ellipse area $\bar{A} = \sum_i A_i$ across target tilts where $A_i$ is the elliptical area associated with each groundtruth tilt $\tau_i$. For a given average equivalent diameter, the areas of the adaptive elliptical pooling regions across different target tilts are proportional to the areas of the ellipses fit to the natural scene statistics (see Figs 5C and 9A and 9B).

To enable direct comparison of the two global pooling models, the average equivalent diameter of the adaptive pooling model is matched with the diameter of a fixed circular pooling region. Then, estimation and prediction errors from the two models are plotted against each other as a function of average equivalent diameter (Fig 9C). Adaptive elliptical pooling causes a small but robust improvement in estimation performance (Fig 9C; blue curve); the minimum estimation errors from adaptive elliptical pooling were lower than those errors from fixed circular pooling on 1000 randomly sampled sets of stimuli (inset in Fig 9C).

The improvement of overall estimation performance by adaptive elliptical pooling leaves open the possibility that adaptive pooling produces a large benefit at one or only a small number of target tilts while hurting performance at other target tilts. If the natural scene statistics indeed govern the pooling rules that optimize performance, a performance improvement should be observed at each target groundtruth tilt. To check, we examined the performance of the fixed circular pooling vs. adaptive elliptical pooling at each target tilt (Fig 9D; S4 Fig). Adaptive elliptical pooling outperforms fixed circular pooling at all target tilts. The results indicate that adaptive elliptical pooling improves performance compared to fixed circular pooling at each individual target tilt and provides further evidence that pooling rules governed by natural scene statistics improve estimation performance.

The story is a bit different when it comes to human prediction error. Adaptive elliptical pooling and fixed circular pooling provide equivalently good predictions of human tilt estimation (Fig 10); similar patterns of performance are obtained for individual human observers (S1 Fig). Human prediction error therefore does not allow us to discriminate between fixed circular and adaptive elliptical pooling in so far as their ability to predict human performance. To determine which of the two models provides a better account of human performance, additional analyses are necessary.

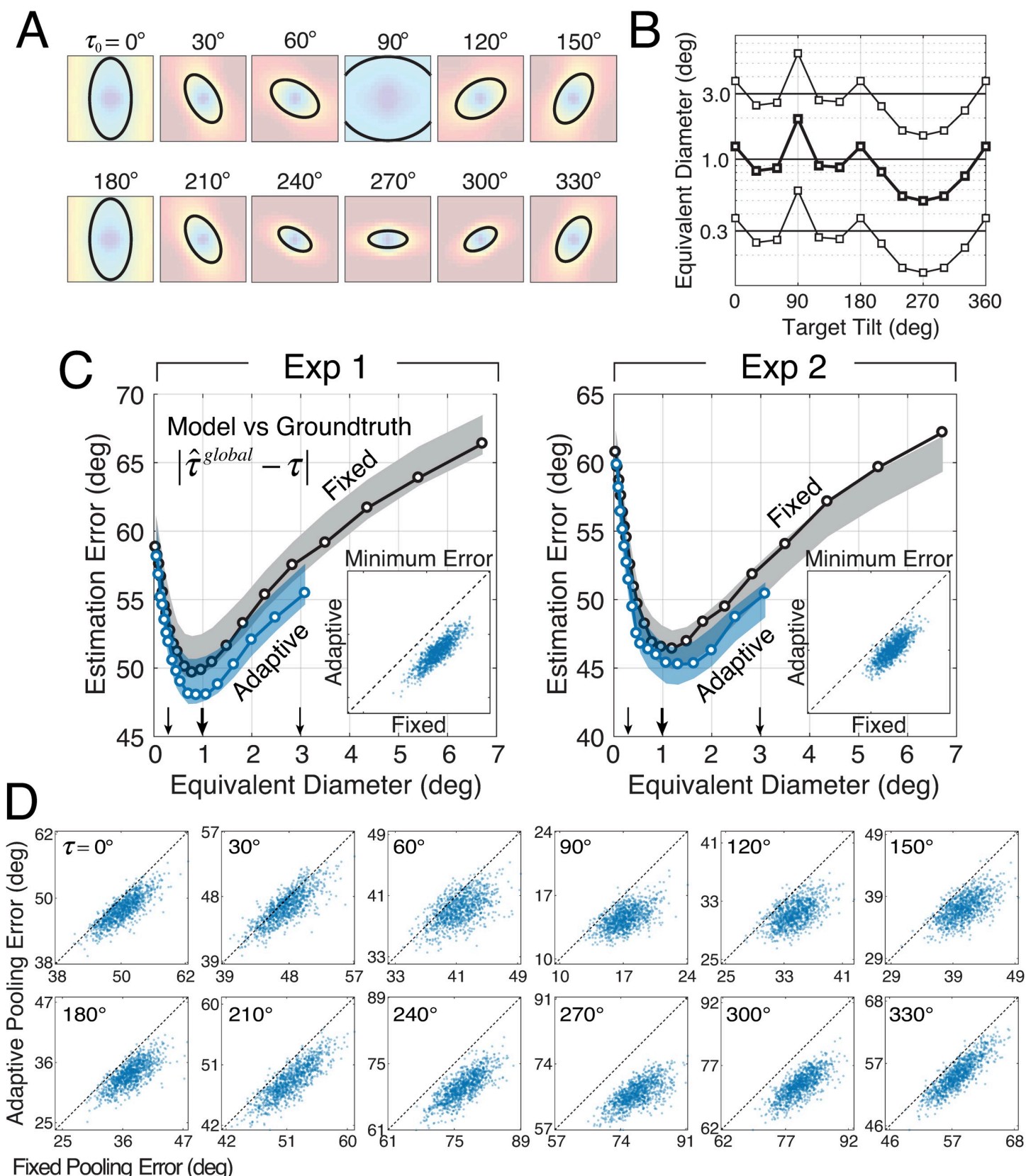

**Fig 9. Estimation error of adaptive elliptical pooling model. A** The adaptive elliptical pooling areas dictated by target tilt. **B** The relative elliptical pooling area for different target tilts. As the average equivalent diameter increases or decreases, the relative sizes of the pooling area remain in a fixed proportion. **C** Estimation error (model estimate vs. groundtruth tilt) as a function of equivalent diameter. The insets show simulation results that compare performance of the adaptive elliptical pooling model vs. the fixed circular pooling model on 1000 matched randomly sampled stimulus sets. Computing the estimation errors on matched stimulus sets isolates the impact of the model, and prevents stimulus variability from unduly affecting the results. The adaptive pooling model (blue) outperforms the fixed circular pooling model (black) on nearly all stimulus sets (i.e., data is below positive diagonal). **D** Simulation results, just as in C insets, except that estimation error is shown as a function of groundtruth tilt (subpanels). The fact that the majority of points lie below the dashed unity line, indicates that adaptive elliptical pooling outperforms fixed circular pooling in groundtruth tilt estimation at all groundtruth tilts.

To further examine whether adaptive or fixed pooling provides a better account of human visual processing, we determined the pooling region size that best accounts for human performance at each target tilt. If the fixed pooling model is the best account of human performance, human pooling at all target tilts should be best accounted for by similarly sized pooling regions. Otherwise, the size of the best pooling region should vary systematically with the tilt at the target location. In the analyses presented thus far, the areas used in the adaptive elliptical pooling model to estimate groundtruth tilt and predict human performance were fixed by fits to the natural scene statistics (Figs 2E,5C,9A and 9B). But the areas determined by these fits do not necessarily match the areas that maximize the accuracy of groundtruth tilt estimation or the prediction of human performance. Thus, we independently determined the size of the elliptical pooling region that maximizes performance (i.e., minimizes error) at each groundtruth tilt. Fig 11A and 11B plot the best equivalent pooling diameters for estimation at each groundtruth tilt against the equivalent pooling diameters that were fit to the natural scene statistics. The correlation is strong. Fig 11C and 11D plot the pooling diameters that best predict human performance at each groundtruth tilt. The same conclusions hold. Similar results are obtained if the natural scene statistics are fit over a larger spatial area (S5 Fig). Thus, the best

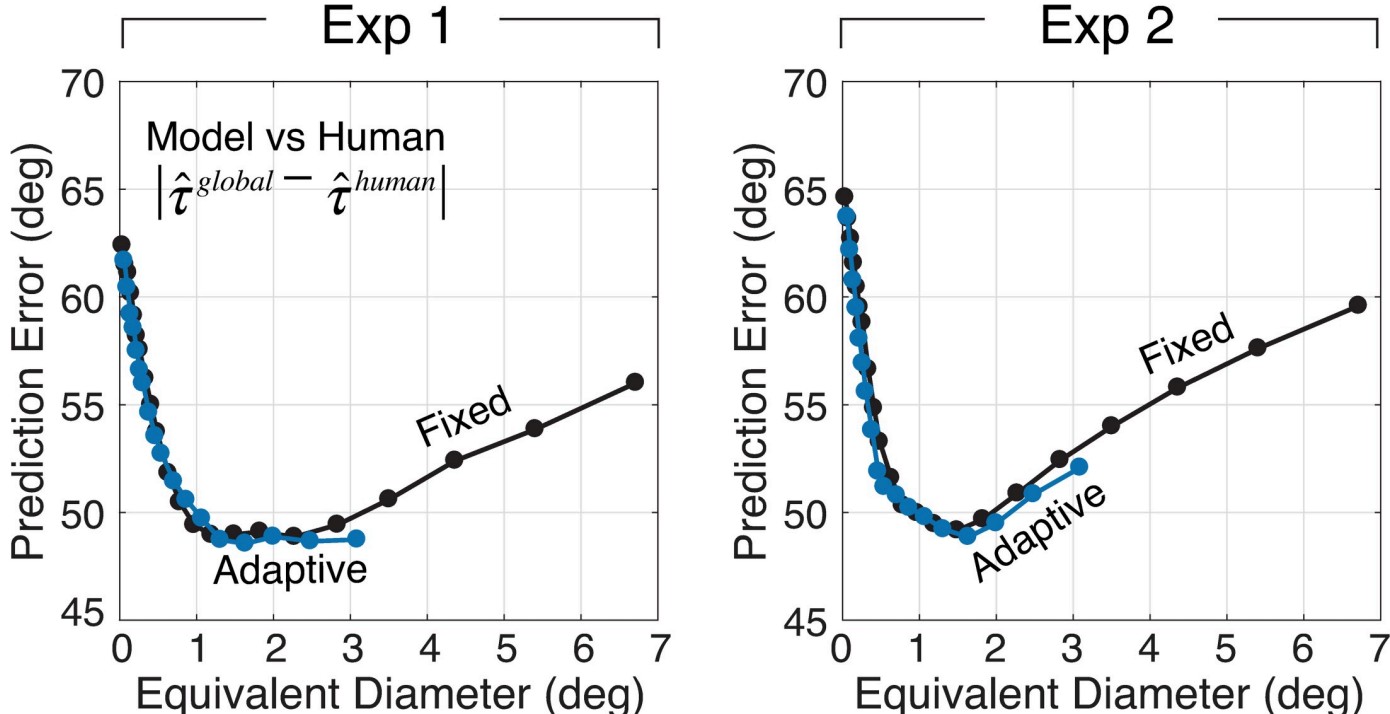

**Fig 10. Prediction error of adaptive elliptical pooling model.** Human prediction error (model estimate vs. human estimate) is plotted (blue) as a function of pooling area (i.e., equivalent diameter). For comparative purposes, performance is also plotted for the fixed circular pooling model (black; same data as Fig 8).

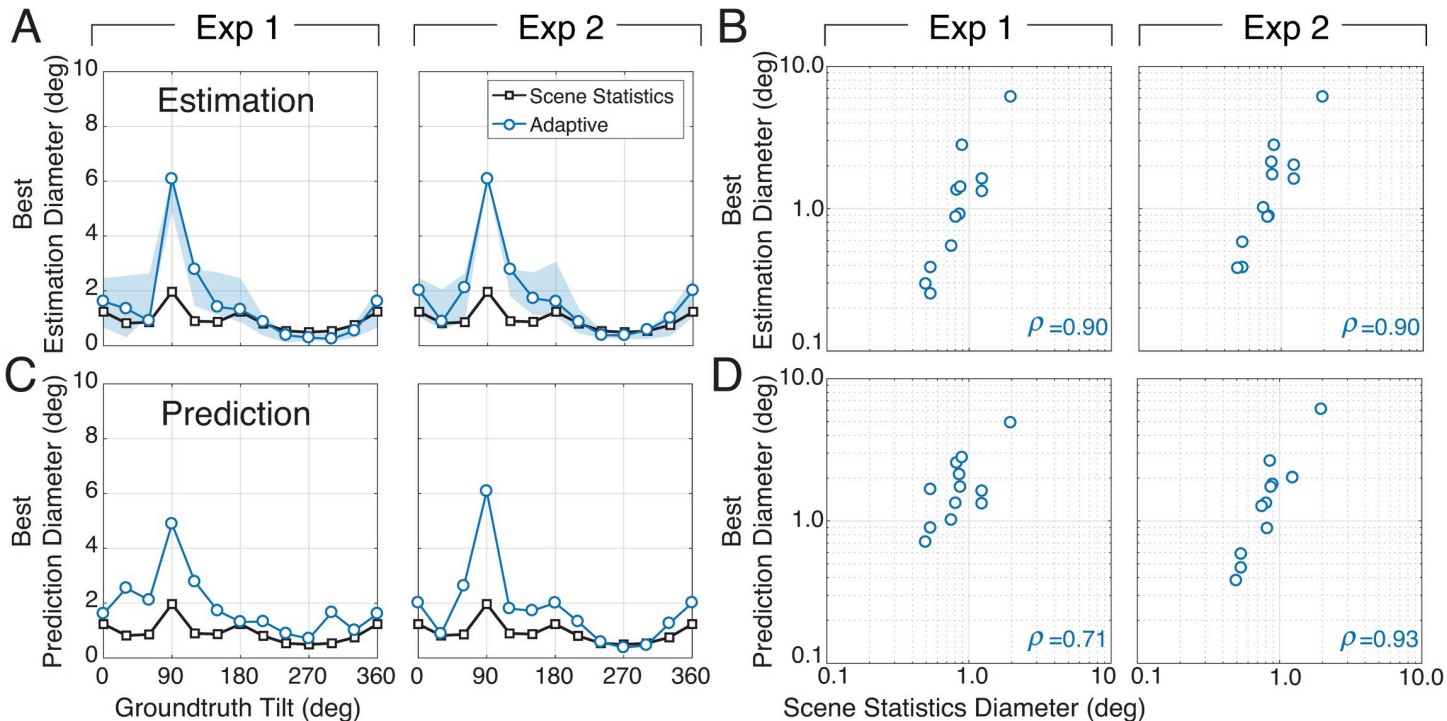

**Fig 11. Pooling sizes from natural scene statistics vs. pooling sizes that maximize estimation and prediction performance.** Adaptive pooling regions predicted by natural scene statistics predict the pooling regions that independently maximize performance at each groundtruth tilt. **A** Equivalent pooling diameters fit to the natural scene statistics (black; same data as Fig 9B) and equivalent pooling diameters that minimize estimation error are plotted as a function of groundtruth tilt. The left and right columns represent data from Exp 1 and Exp 2, respectively. **B** Best estimation diameters are correlated with the diameters fit to the natural scene statistics. **C** Equivalent pooling diameters fit to the natural scene statistics (black) and equivalent pooling diameters that minimize prediction error (blue), plotted as a function of groundtruth tilt. **D** Best prediction diameters are correlated with the diameters fit to the natural scene statistics. All correlations were significant at the level of p<0.05; all but one were significant at the level of p<0.001.

pooling diameters for estimating groundtruth tilt and predicting human performance are tightly correlated with those obtained by fits to the natural scene statistics. These results favor the adaptive elliptical pooling model over the fixed circular pooling model as the best account of human visual processing, given that the fixed circular pooling strategy predicts no change in pooling diameter with groundtruth tilt. Natural scene statistics therefore provide a solid prediction for how signals should be pooled across space to maximize the estimation of groundtruth tilt and the prediction of human performance.

Finally, we compared the sizes of the best pooling diameters (cf. Fig 11) for groundtruth tilt estimation against those that are best for predicting human performance. In some sense, this is the most direct test of the hypothesis that natural scene statistics guide how humans pool information across space in surface tilt estimation. If humans use the pooling regions that yield the most accurate performance, humans are appropriately using the information available in natural images to perform the task. The sizes of the adaptive pooling regions that maximize estimation and prediction performance are strongly related to one another (Fig 12). In fact, when the size of the adaptive pooling region is based on the local tilt estimate at the target location, the best estimation diameters (i.e., the equivalent pooling diameters that maximize estimation performance) are nearly the same as the best prediction diameters (i.e., the pooling diameters that maximize the model ability to predict human performance).

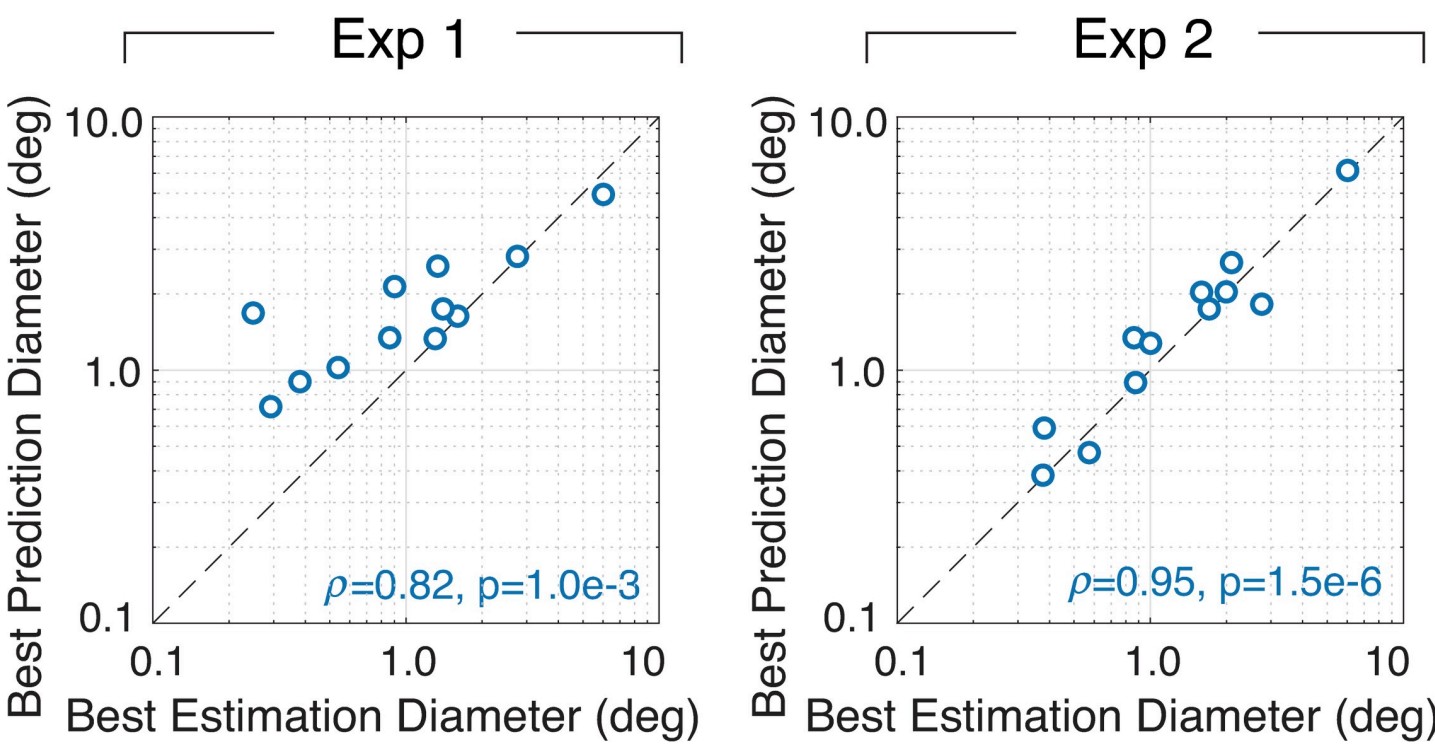

**Fig 12. Pooling sizes that maximize groundtruth tilt estimation performance vs. pooling sizes that maximize the prediction of human performance.** Pooling diameters that maximize estimation performance predict those that maximize the prediction of human performance. Each data point represents the diameter that maximizes performance for a different groundtruth tilt at the target location (cf. Fig 11). The actual sizes of the pooling regions that maximize estimation performance are similar to the sizes that maximize the prediction of human performance.

## Discussion

In this paper, we analyzed the spatial statistics of surface tilt in natural scenes and found that a hierarchical model of surface tilt estimation that pools local tilt estimates according to these statistics provides more accurate estimates of groundtruth tilt and better predictions of human tilt estimation than a principled model that bases performance only on local image cues [15,16]. Additionally, we have shown that the spatial scale of the pooling region that maximizes groundtruth tilt estimation performance is similar to the spatial scale that optimizes the model ability to predict human performance. Together, these findings show that natural scene statistics predict how humans pool information across space in surface tilt estimation.

### The evidence for pooling

The current results show that a two-stage hierarchical model of visual information processing provides a good account of tilt estimation in natural scenes. To be confident that the second (i.e., pooling) stage of the model is necessary, the performance improvements of the global model should be due at least in part to the explicit pooling of local estimates, and not due simply to the fact that the global model uses image information from a larger spatial area than the local model (cf. Fig 4C). To examine this issue, we determined whether the performance of a local (i.e., one-stage) model that uses information from the same area of the image as the global model can equal the ability of the global model to estimate groundtruth tilt and predict human performance. An area-matched "local" model, which computes cue values from an area of 1.0˚ closely approximates the image area from which the best global model implicitly uses image

information. (The image area that contributes to each global estimate is the sum of the areas of the pooling region and the Gaussian derivative operator used to compute the local image cues; see Methods.) This area-matched local model underperformed the best global model, both in its ability to estimate groundtruth tilt and its ability to predict human performance (see Figs 7 and 8, and S3 Fig).

Another way to put the performance of the global model in context is to compare the amount by which it reduces prediction error given what is possible due to the agreement of the human observers' estimates. To do so, we constructed a toy model that outputs a tilt estimate equal to the circular mean of the human estimates to each stimulus. This toy model provides a best-case-scenario in terms of its ability to minimize human prediction error (c.f. Fig 8), but it is a wholly implausible working model of tilt estimation because it operates only post-hoc on the human responses and not on the image data. Nevertheless, across all stimuli in both experiments, the mean prediction and estimation errors of the toy model represent 39˚ and 13º improvements over the local model, respectively. The global model, by comparison, achieved improvements in prediction and estimation errors of 15º and 14º, respectively, over the local model, which is notable given that the global model used only natural image data as inputs.

Note that we cannot exhaustively examine all possible local models. We therefore cannot rule out the possibility that there exists some local model—for example, that uses a different set of local cues—that can achieve performance equivalent to the global model. However, at least in the space of models that we have considered, the demonstrated benefit of global pooling cannot be trivially explained by the fact that the global model implicitly uses image information over a larger area than the local model. The demonstrated links between the scene statistics and the best performing adaptive global model suggest that pooling according to the natural scene statistics benefits performance.

## Two psychophysical experiments

We measured human tilt estimation performance in two experiments. Experiment 1 sampled surfaces with a larger range of slants (ranging between 30º and 75º) than Experiment 2 (slants ranging between 30º and 60º). In classical psychophysical experiments on surface orientation, it is typical for the projected size of a stimulus on the retina to be held constant regardless of the surface slant. In these circumstances, slant (and presumably tilt) discrimination thresholds decrease as slant increases [2–4,35]. For surfaces of a given size, however, the projected size of the stimulus decreases as slant increases. For very large slants, the smaller area of the projection should make the estimation of slant and tilt more difficult, especially when the surfaces are embedded in a cluttered environment as is often the case in natural scenes. In these circumstances, the estimation of groundtruth tilt may also be adversely affected by nearby depth discontinuities. As it turns out, there was a small improvement in the performance in the experiment with the more restricted range of slants (i.e., Exp 2), but the overall patterns of performance were quite similar across both experiments. A more targeted set of experiments would be necessary to rigorously examine these particular issues.

## Visual systems and the internalization of natural scene statistics

In recent years, a series of papers have provided evidence linking certain statistical aspects of natural images and scenes [15,28–30,36–39] to the design of the human visual system [40–42], and to the performance of ideal and human observers in perceptual tasks [14,16,38,43–52]. This broad program of research has, with varying degrees of rigor, invoked natural scene statistics to account for a strikingly diverse set of topics: how the shape of pupils changes across species in different ecological niches [42], where corresponding points are located in the two

retinas [40,41], how biases in binocular eye movements manifest [49], how targets are detected in natural images [48], how image contours are perceptually grouped [38,43], how image orientation is estimated [46], how focus error is estimated [50,51], how binocular disparity is estimated [45,53,54], how image motion is estimated [47,52,55], how 3D tilt is estimated [16], and now, how cues to 3D tilt are pooled across space. Over this same period, numerous modeling frameworks have emerged that provide theoretical and computational methods for predicting and accounting for these links [53,56–59]. The coming years are likely to demonstrate more links between properties of natural scenes and functional properties of sensory-perceptual systems.

## Adaptive spatial spooling

The current work indicates that the human visual system adaptively pools information across a spatial neighborhood that is closely related to the spatial neighborhood that maximizes the ability to estimate groundtruth tilt in natural scenes. A number of investigations have found evidence for adaptive spatial pooling. Local image properties (e.g., contrast) at a target location influence the spatial region over which information is integrated, both at the level of individual neurons and at the level of perception [60,61]. The current work shows that, in the domain of surface tilt estimation, the rules governing adaptive pooling are linked to the statistics of natural scenes. However, local estimates should only be pooled if they carry information about the same physical source; local estimates should not typically be pooled across depth boundaries [62–64]. The current work is limited in that it does not explicitly address how the visual system should avoid pooling across depth boundaries. This is left as a project for future work.

## The definition of groundtruth tilt

To perform this study, it was necessary to operationalize the definition of groundtruth surface orientation (i.e., slant and tilt). Doing so requires choosing a spatial scale at which to evaluate small local changes in viewing distance, as it is these local changes that are required to quantify slant and tilt. For surfaces with local 3D texture, groundtruth slant and tilt will tent to vary with the spatial scale at which these local changes are evaluated. This is a critical difference between many real-world surfaces and the planar surfaces that are typically used in laboratory experiments to probe the visual system. Another complicating factor is that the range data from which groundtruth surface orientation was computed, was measured with a LIDAR sensor. The measurement resolution determines the smallest spatial scale at which surface orientation can be computed. In this study, we chose to define groundtruth tilt at the smallest spatial scale that was practical to compute from the LIDAR samples (see Methods). There is no guarantee that the human visual system aims to recover the orientation of surfaces in scenes at this particular spatial scale, or any single spatial scale. The consistency between the adaptive global pooling model and human performance (see Figs 10–12) implies that the human visual system adaptively picks the spatial scale with which to use image information in tilt estimation. Does this result imply that the spatial scale at which groundtruth tilt is estimated adapts with the viewing situation? This is a difficult question and one we are not yet prepared to answer. Future work on surface orientation estimation in natural scenes will have to grapple with how to do so. Similar issues will arise for other estimation problems (e.g., shape, length) where the groundtruth property of interest varies locally and spatial scale is a fundamental determinant of the groundtruth property to be estimated [65].

## Spatial pooling and cue combination

The logic underlying the spatial pooling rules investigated here is closely related to the logic underlying standard theories of cue combination. Spatial pooling and cue combination both

rely on the simple fact that multiple sources of information are better than one, provided that the sources are properly combined. In the current analysis of spatial pooling, the individual local tilt estimates play a similar role that individual cues play in cue combination. The difference between cue combination and spatial pooling is in the point of emphasis. Cue combination has most often been studied for cases in which multiple measurements of a single groundtruth value are available at the same spatial location (but see [33,66,67]). Spatial pooling, by contrast, focuses on the integration of pieces of information at multiple different spatial locations, which often correspond to multiple groundtruth values. Developing computational and experimental paradigms to rigorously explore these distinctions is an important goal for future work.

## Methods

### Human observers

Four human observers participated in the two experiments; two authors and one naïve subject participated in Experiment 1, and one author and a different naïve subject participated in Experiment 2. Informed consent was obtained from participants before the experiment. The research protocol was approved by the Institutional Review Board (IRB) at the University of Pennsylvania (protocol number 824435) and is in accordance with the Declaration of Helsinki.

### Natural scene database

Natural stimuli were sampled from a recently published natural scene database containing stereo-images with precisely co-registered distance data of natural scenes [15]. The images for left and right eyes were taken at two positions separated by a typical human inter-pupillary distance (6.5cm). Scenes were photographed such that no objects were nearer than 3m and such that all images were sharp. The left- and right-eye images associated with each of the 95 stereo pairs have a resolution of 1080x1920 pixels. The natural scenes depicted in the database contain buildings, streets, shrubs, trees, and open green areas.

### Apparatus

Stereo-image patches were presented with a ViewPixx Technologies ProPixx projector fitted with dynamic polarization filter. Left- and right-eye images were temporally interleaved at a refresh rate of 120Hz (60Hz per eye). Projected images were displayed on a 2.0x1.2m Harkness Clarus 140XC polarization-maintaining screen and viewed through passive polarization maintaining goggles. At the 3m viewing distance, the screen subtended 36°x21° of visual angle. The 3m screen distance minimized screen cues to flatness depth because the blur detection threshold is approximately 1/3D [68]. Head position was stabilized by a chinrest and a headrest. The display system nearly recreates the retinal images that would have been formed by the original scenes. The primary difference is that the overall intensity of the light reaching the eyes is lower because sunlight is more intense than the max intensity produced by the projector (84cd/m$^2$).

### Experimental stimuli

Each natural scene was viewed binocularly in gray scale through a virtual stereoscopic aperture. The aperture had a diameter of 3˚ of visual angle and was positioned 5arcmin of disparity in front of the surface that appeared at its center (Fig 3A). Scene locations (i.e., patches) were sampled with a number of constraints. In Experiment 1, patches were excluded i) if the center

pixel was associated with a surface slant of less than 30º or more than 75º, (ii) if the center pixel was associated with a surface distance that was less than 5m or larger than 50m, (iii) if the center pixel was in a half-occluded region, and (iv) if the root-mean-squared contrast of the patch was less than 5% or greater than 40%. In Experiment 2, all constraints were the same except the acceptable surface slants were between 30º and 60º instead of between 30º and 75º. Stimuli were selected so that all tilts were evenly represented in the experiments. For each of 24 bins that were 15º wide (24 bins x 15º = 360º), 150 stereo-image patches were selected for a total of 3600 unique patches (3600 = 24 x 150). In both experiments, the patches were displayed at the image location from which they were sampled.

## Procedure

Data was collected in 24 blocks of randomly permutated trials. Each block consisted of 150 trials and lasted approximately 10min. On each trial, observers estimated the tilt at the center of each patch. The task was to match the surface tilt angle with the orientation of a mouse-controlled graphical probe (Fig 3B). The initial probe orientation was randomly selected. There was no time limit for the response. No feedback was provided.

## Groundtruth tilt

Groundtruth tilt $\tau$ is computed from the distance data contained in the range map **r** that is co-registered to each natural image in the database. We defined groundtruth tilt arctan($\nabla_y$**r**,$\nabla_x$**r**) as the orientation of the range gradient [1], where arctan($\cdot$) is the four quadrant arctangent function. The range gradient was computed by first convolving the groundtruth distance data with a 2D Gaussian having a space constant $\sigma_{tilt}$ of 3arcmin and then taking the partial derivatives in the $x$ and $y$ image directions [15,16]. A space constant of this size corresponds to an image area of approximately 0.25ºx0.25º. These operations are mathematically equivalent to convolving the distance data with $x$ and $y$ Gaussian derivative operators with the same space constant.

## Image cues to tilt

Image cues were computed directly from the images. The disparity and luminance cues were defined as the orientation of the local disparity and luminance gradients arctan($\nabla_y cue$,$\nabla_x cue$) and were computed by convolving the cue image with a 2D Gaussian having space constant $\sigma_{cue}$ of 6arcmin and then taking the partial derivatives in the $x$ and $y$ image directions. The disparity image was computed from the left- and right-eye luminance images via standard local windowed cross-correlation [15,31,69]. The cross-correlation window had the same space constant as the derivative operator that computed the gradient. The texture cue was defined as the dominant orientation of the major axis of the local amplitude spectrum [10,15]. The unsigned cue values were obtained by taking the 180º modulus of the signed cues.

## Fitting elliptical pooling regions to scene statistics

To determine how 3D surface tilt is spatially related in natural scenes we computed the absolute mean tilt difference as a function of spatial offset in the image (Fig 2D and 2E). Then, we fit a 2D Gaussian to the map of tilt differences after scaling the map so the volume equaled 1.0. The aspect ratio and relative size of the elliptical pooling regions were determined from the covariance matrix of the best-fit Gaussian, the iso-level curves of which are ellipses. The orientation of the elliptical pooling regions was aligned with the orientation of the target tilt.

## Mean absolute tilt difference

Tilt is a circular (i.e., angular) variable. The difference between two tilts is determined via the circular distance, the standard method of computing the difference between circular variables. The absolute circular distance between a pair of tilts at two different locations is given by

$$|\tau_i - \tau_0| = |\arg(\exp[j(\tau_i - \tau_0)])|, \tag{5}$$

where $\tau_0$ is the tilt at the target location, $\tau_i$ is the tilt at a neighboring location, and $j = \sqrt{-1}$ is the imaginary unit number. The mean circular distance across pairs of tilts in a given spatial relationship is given by

$$E[|\tau_i - \tau_0|] = \arg(\sum_{k=1}^{N} \exp(j|\tau_{i_k} - \tau_{0_k}|)), \tag{6}$$

where $N$ is the number of tilt pairs contributing to the mean. The mean tilt differences (i.e., circular means) are plotted as a function of spatial location (relative to a target location) in Fig 2D and 2E.

## Local model: Estimating tilt magnitude

Tilt magnitude (i.e., unsigned tilt) is estimated from three *unsigned* tilt cues, $\mathbf{C}_u = [L_u, T_u, D_u]$, where $L_u$ is the unsigned luminance cue, $T_u$ is the unsigned texture cue, and $D_u$ is the unsigned disparity cue at the target location. Each cue is quantized (i.e., binned) into 64 discrete values. Thus, across all three cues, there were $64^3$ unique bins. Analyzes showed that this binning process produced a negligible effect on estimation performance. The tilt estimate is the conditional mean $\hat{\tau}_u = E[\tau_u | \mathbf{C}_u]$ given a triplet of image cue measurements $\mathbf{C}_u$ (Fig 4). The conditional mean is identical to the mean of the posterior over unsigned tilt assuming a minimum circular distance cost function (i.e., analogous to the mean-squared-error cost function for linear variables). The posterior mean equals the sample mean from a large number of samples of $\tau_u$ in the natural scene database, assuming the samples are representative. Tilt is a circular variable. The conditional mean is thus given by

$$\hat{\tau}_u = E[\tau_u | \mathbf{C}_u] = \arg(\sum_{\tau_u \in \Omega} e^{j\tau_u}), \tag{7}$$

where $\Omega_u$ is the set of unsigned groundtruth tilts $\tau_u$ co-occurring with the triplet of cue values $\mathbf{C}_u$. On test images, the cue triplet is computed from the images and the optimal tilt is obtained from a lookup table (cf. estimate cube in Fig 4A).

## Local model: Estimating tilt sign

Tilt sign is determined from the *signed* disparity cue only $\mathbf{C}_s = [D_s]$, where $D_s$ is the signed disparity cue at the target location. The signed disparity cue was quantized into 64 bins. The sign of tilt is computed from the conditional mean of signed tilt given the signed cue.

$$\text{sgn}(\hat{\tau}_s) = \text{sgn}(E[\tau_s | \mathbf{C}_s]) = \text{sgn}(\arg(\sum_{\tau_s \in \Omega} e^{j\tau_s})), \tag{8}$$

where $\Omega_s$ is the set of signed groundtruth tilts co-occurring with the signed disparity values $\mathbf{C}_s$.

## Supporting information

**S1 Fig. Experiment 1 estimates from individual human observers and models. A** Histogram of raw responses. **B** Mean of tilt estimates as a function of groundtruth tilt (binned in 24 bins). **C** Variance of tilt estimates as a function of groundtruth tilt. Data from the fixed circular pooling model corresponds to the best pooling area (i.e., 1º diameter). Data from the adaptive elliptical model corresponds to the adaptive model with the best average pooling area (i.e., 1º average equivalent diameter). The variances of the human tilt estimates are substantially more similar to the variances of the tilt estimates from the global pooling models (fixed & adaptive) than from the local model.
(TIF)

**S2 Fig. Experiment 2 estimates from individual human observers and models. A** Histogram of raw responses. **B** Mean of tilt estimates as a function of groundtruth tilt (binned in 24 bins). **C** Variance of tilt estimates as a function of groundtruth tilt. Data from the fixed circular pooling model corresponds to the best pooling area (i.e., 1º diameter). Data from the adaptive elliptical model corresponds to the adaptive model with the best average pooling area (i.e., 1º average equivalent diameter). The variances of the human tilt estimates are substantially more similar to the variances of the tilt estimates from the global pooling models (fixed & adaptive) than from the local model. The variances of the human and model tilt estimates in Exp 2 exhibit substantially different patterns than in Exp 1. The differences are due to the different sets of natural stimuli that were presented during the experiment.
(TIF)

**S3 Fig. Prediction error for individual human observers.** Prediction error is shown for the fixed circular pooling model (black), the adaptive elliptical pooling model (blue). The top and bottom rows indicate results from Exp 1 and Exp 2, respectively. The pooling region that minimizes prediction error for all models and all human observers (except observer S3) corresponds to an equivalent pooling diameter between 1º and 2º. The black dashed line indicates the prediction error for the local model. The gray dashed line indicates the prediction error for a "local" model that computes the image cues from an area matched to that implicitly used by the best global model.
(TIF)

**S4 Fig. Estimation error with fixed circular vs. adaptive elliptical pooling for different groundtruth tilts in Experiment 2.** Each point represents the mean estimation error in a randomly sampled stimulus set across stimuli at a given groundtruth tilt. Estimation error with fixed circular pooling is plotted against estimation error with adaptive elliptical pooling. Computing the prediction errors on matched stimulus sets isolates the impact of the model, and prevents stimulus variability from unduly affecting the results. The fact that the majority of points lie below the dashed unity line, indicating that adaptive elliptical pooling outperforms fixed circular pooling for the task of estimating groundtruth tilt in natural scenes.
(TIF)

**S5 Fig. Robustness of natural scene statistics predictions. A** Spatial statistics of tilt in natural scenes over a 1deg area. Mean absolute tilt difference as a function of spatial location relative to a target location. **B** Mean absolute tilt difference conditioned on the groundtruth tilt at the target location. **C** Fits to the scene statistics in B. **D** Equivalent diameters of the fits to the scene statistics in C. **E** Adaptive pooling regions predicted by natural scene statistics predict the pooling regions that maximize performance at each groundtruth tilt. Equivalent pooling diameters fit to the natural scene statistics (black) and equivalent pooling diameters that minimize

estimation error (blue), plotted as a function of groundtruth tilt. The left and right columns represent data from Exp 1 and Exp 2, respectively. **F** Best estimation diameters are correlated with the diameters fit to the natural scene statistics. **G** Equivalent pooling diameter fit to the natural scene statistics and equivalent pooling diameters that minimize prediction error, plotted as a function of groundtruth tilt. **H** Best prediction diameters are correlated with the diameters fit to the natural scene statistics.
(TIF)

## Author Contributions

**Conceptualization:** Seha Kim, Johannes Burge.

**Data curation:** Seha Kim.

**Formal analysis:** Seha Kim, Johannes Burge.

**Funding acquisition:** Johannes Burge.

**Investigation:** Seha Kim, Johannes Burge.

**Methodology:** Seha Kim, Johannes Burge.

**Resources:** Johannes Burge.

**Software:** Seha Kim, Johannes Burge.

**Supervision:** Johannes Burge.

**Validation:** Seha Kim.

**Visualization:** Seha Kim, Johannes Burge.

**Writing – original draft:** Seha Kim, Johannes Burge.

**Writing – review & editing:** Seha Kim, Johannes Burge.

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
