## [Decision Letter · Decision Letter 0]

2 Mar 2020

Dear Dr Kim,

Thank you very much for submitting your manuscript "Natural scene statistics predict how humans pool information across space in surface tilt estimation" for consideration at PLOS Computational Biology.

As with all papers reviewed by the journal, your manuscript was reviewed by members of the editorial board and by several independent reviewers. In light of the reviews (below this email), we would like to invite the resubmission of a significantly-revised version that takes into account the reviewers' comments.

We cannot make any decision about publication until we have seen the revised manuscript and your response to the reviewers' comments. Your revised manuscript is also likely to be sent to reviewers for further evaluation.

Sincerely,

Leyla Isik

Associate Editor

PLOS Computational Biology

Wolfgang Einhäuser

Deputy Editor

PLOS Computational Biology

Reviewer's Responses to Questions

**Comments to the Authors:**

Reviewer #1: Review of "Natural scene statistics predict how humans pool information across space in surfact tilt estimation"

Reviewer: Michael Landy

This paper describes experimental data on estimation of surface tilt from patches of stereo images from a natural-image database that is coupled with range data, so that ground-truth tilt is known. They find that adding spatial context (local estimates from surrounding pixels) improves the performance of their computational observer and also improves the match of predicted responses with human responses, suggesting that human tilt estimation also uses surrounding context to improve estimates and make them more robust.

This is a nice paper, clearly written and with clear results. I suppose my main overall reactions are twofold. First, This is a follow-up to their previous paper (Ref. 16) and yet this relationship is only made clear in a single sentence (bottom of p. 6). This should be discussed in the introduction, so that it is made clear what the contribution of this paper is above and beyond the previous paper. Also, as I'll mention below, there are spots where knowledge of the methods of the previous paper are required to make sense of the current one, and this one should be able to be read on its own. Second, there really isn't a clear description of how good the improvement of prediction with context is, compared to how good an improvement one might expect. It's a decrease in prediction error of 10 deg or so, but is there an independent measure of estimation variance, so that one can get an absolute sense of whether that's all the improvement any model can manage or, contrarily, whether that's actually fairly weak evidence?

Specifics (page/para/line):

4/3: The slant/tilt of a surface is really a multi-scale phenomenon, at least for surfaces that have local 3-d texture. Thus, the estimate from the database will be a function of the resolution of the measurements (in surface coordinates, i.e., the scale will also be affected by distance to the camera). I don't know whether that's worth a mention/discussion.

Overall: You never really show any raw data (i.e., the distributions of estimates conditioned on the ground-truth value). These are presumably in the other paper, but a reader might be curious about whether there are a lot of tilt-sign errors and, as mentioned above, the amount of variance.

Fig. 3: I have no idea what a "mouse-controlled graphical probe" is, and Fig. 3A certainly doesn't help! ;^)

Eq, 2: This is elaborate, but isn't this just going to turn out to be the sign of the signed disparity cue (i.e., expectation over the database isn't going to change that estimate)?

11/1/4: "the human estimate": You never say what this is (i.e., presumably a circular average across subjects, I guess).

Fig. 9D: The 90 deg data show an interesting phenomenon that isn't discussed (it's also in Fig. S2): fixed closer to adaptive for local-tilt at 90 deg but farther at 270. I'm not sure what, if anything, that means. But, it makes me wonder about the lack of any discussion of tilt priors (preference for cardinals, preference for ground plane over ceiling plane in this case). Sure, experimentally you impose a flat prior, but that doesn't mean the observers ignored their priors.

Many places: You spend a bunch of the paper comparing a model that chooses the pooling region/weights based on the ground-truth local tilt vs. the local estimate. Yet, you never once mention that the former is nonsensical as a model of human (or any) observer behavior, since ground truth is unknown to the observer. I don't really get the point of even including the ground-truth version of the model, given its logical circularity.

Fig. 11B/D: These are correlated, but not slope 1, not identity line, and this is log-log, so the suggestion is that the two don't match and are related by a power function. Huh?

15/1/14: Fig. 11B -> Fig. 11C

19/1: How many trials per block?

19/2/2: A four-quadrant arctan can't be a function of the ratio, but needs the individual values (as it does in Matlab!).

19/2/5: Are the partial derivatives simply first differences in x and y (of neighboring pixels) or what?

19/3/7: "the derivative operator": again, I'm not sure exactly what precise computation is being referenced by this.

19/4/1: "the mean tilt difference": Don't you mean the mean ABSOLUTE tilt difference here?

20/2/4-6: Why are we talking about a "posterior" when we've never talked about a prior? Are you considering the database to be a prior? Or the subsampled one designed to be flat over tilt?

Eqs. 7 & 8: First, since you are computing the arg(), the 1/N is not needed. Second, you should clarify that you aren't conditioning on the exact values of the three cues (c_u), but rather on the bin (of the 64x64x64 binning, that you never clearly describe in this paper, and should).

Reviewer #2: This is an elegant and detailed study of the adaptive pooling of local cues to surface tilt across stereoscopic, luminance and texture cues. The authors use a large, high-fidelity dataset of co-registered image and depth data to measure the spatial statistics of surface orientation in natural scenes, and use this to determine the optimal pooling of signals to improve tilt estimation. They find that the optimal pooling patterns are anisotropic, such that for a given tilt at a given image location, estimation can be improved by pooling signals over elliptical neighbourhoods, favouring directions aligned with the projected tangent plane. They then compare the predictions of the optimal model with human tilt estimation performance, to test the hypothesis that the human visual system also optimally pools local tilt estimates. They consider both isotropic and anisotropic pooling models. The findings show convincingly that the human visual system does pool local signals, and the analyses in Figures 11 and 12 suggest that the visual system does so in an adaptive way that is consistent with the statistics of natural images (at least in terms of pooling region size).

I don’t have major comments, but I do have a number of minor issues and suggestions.

The authors should say more about the difference between Experiments 1 and 2. This includes on p.6: the main text should indicate more clearly that there are two separate psychophysical experiments, and should briefly explain the difference between them. Otherwise, references to Experiments 1 and 2 are confusing throughout the rest of the MS. More importantly, a subsection should be added to the discussion section, discussing the difference in prediction quality for the two experiments. Is the problem with higher slants (Experiment 1) the presence of artifacts in the image set? Or is the fact that higher slants tend to be associated with depth discontinuities within the pooling region? Were the results of Experiment 1 evaluated before deciding to run Experiment 2? Some rationale and discussion would be helpful.

It would also be helpful to be given stronger intuitions about how local estimates are affected by estimates that come from outside the pooling region. Each pixel within the pooling region for a given local estimate is itself adjusted by its own pooling region. Does this ‘cascading’ extend to the whole image? If not, why not? How quickly does the effect dilute as a function of distance outside the pooling region? The authors could add an analysis to quantify this (it should not be so difficult to do: values can be clamped to the local estimates vs. all values modified by their own pooling regions).

It would also be useful to understand the relative contribution of the three different cues in the local estimates. How far off is the disparity estimate from the nominal ground truth? Does this boil down to basically a study of how to compute optimal derivatives of depth-from-stereo estimates? A few sentences on this would be useful.

Why are the estimated and prediction errors so high? 60 degrees error for the local estimates seems quite large, especially considering that chance is 90 degrees. That means that tilt estimates are closer to random than to the ground truth. Or have I misunderstood something?

Figure 2D: an indication of the marginal frequency of the different tilts would be helpful. For example, 90 deg is presumably significantly more common than others.

An interesting control condition for the experiment would be to rotate the image patches by various amounts so that their local statistics would be the same, but their relationship to the global statistics of tilts in the real world would be changed. Would the best pooling model rotate with the patches or not? This would provide insights into the cues the visual system uses to adapt the pooling region: is it just a prior based on the local tilt estimate, or is it based on recognizing the image content in some way? I don’t think it is necessary to perform this experiment for the paper to be publishable, but it might be interesting to mention it in the discussion section, or, ideally do a small control condition with a smaller number of image patches.

Typos etc:

p. 8 ‘properties of tilt natural scenes’ -> ‘properties of tilt in natural scenes’

Figures 7 and 8: Why is the mean estimate so close to the lower bound of the confidence intervals in Experiment 1 and so close to the top in Experiment 2 (at least above 2deg)?

p. 15: ‘should be best accounted for similarly sized pooling regions’. There is a missing ‘by’.

p. 16: ‘If humans use the pooling regions that yield the most accurate performance, humans are doing the right thing’. Perhaps a less colloquial formulation can be found?

**Have all data underlying the figures and results presented in the manuscript been provided?**

Reviewer #1: No:

Reviewer #2: Yes

PLOS authors have the option to publish the peer review history of their article (what does this mean?). If published, this will include your full peer review and any attached files.

Reviewer #1: Yes: Michael S Landy

Reviewer #2: No
---

## [Decision Letter · Decision Letter 1]

14 May 2020

Dear Dr Kim,

We are pleased to inform you that your manuscript 'Natural scene statistics predict how humans pool information across space in surface tilt estimation' has been provisionally accepted for publication in PLOS Computational Biology.

Best regards,

Leyla Isik

Associate Editor

PLOS Computational Biology

Wolfgang Einhäuser

Deputy Editor

PLOS Computational Biology

Reviewer's Responses to Questions

**Comments to the Authors:**

Reviewer #1: Looks good. All my comments were adequately addressed. One typo/dropout: Methods/Groundtruth tilt/line 6: in the [x] and [y] image directions?

Reviewer #2: The authors have satisfactorily addressed my comments and I think the MS is ready for publication. Lovely work!

**Have all data underlying the figures and results presented in the manuscript been provided?**

Reviewer #1: No: I didn't see any mention of this.

Reviewer #2: Yes

PLOS authors have the option to publish the peer review history of their article (what does this mean?). If published, this will include your full peer review and any attached files.

Reviewer #1: Yes: Michael S Landy

Reviewer #2: No

---

## [Editor Report · Acceptance letter]

11 Jun 2020

PCOMPBIOL-D-19-02169R1 

Natural scene statistics predict how humans pool information across space in surface tilt estimation

Dear Dr Kim,

I am pleased to inform you that your manuscript has been formally accepted for publication in PLOS Computational Biology. Your manuscript is now with our production department and you will be notified of the publication date in due course.

With kind regards,

Laura Mallard
